

# Stomatal conductance bears no correlation with transpiration rate in wheat during their diurnal variation under high air humidity

Xinying Zhang[1,2,3], Xurong Mei[1,2,3], Yajing Wang[1,2,3], Guirong Huang[1,2,3], Fu Feng[1,2,3], Xiaoying Liu[1,2,3], Rui Guo[1,2,3], Fengxue Gu[1,2,3], Xin Hu[4], Ziguang Yang[5], Xiuli Zhong[1,2,3] and Yuzhong Li[1,2,3]

[1] Institute of Environment and Sustainable Development in Agriculture, Chinese Academy of Agricultural Sciences, Beijing, China
[2] State Engineering Laboratory of Efficient Water Use and Disaster Mitigation for Crops, Beijing, China
[3] Key Laboratory for Dryland Agriculture of Ministry of Agriculture, Beijing, China
[4] Institute of Wheat Research, Shangqiu Academy of Agriculture and Forestry Sciences, Shangqiu, China
[5] Luoyang Academy of Agriculture and Forestry, Luoyang, China

Corresponding authors
Xiuli Zhong, zhongxiuli@caas.cn
Yuzhong Li, liyuzhong@caas.cn

## ABSTRACT

A good understanding of the response of photosynthesis rate ($P_N$) and transpiration rate ($Tr$) to stomatal alteration during the diurnal variations is important to cumulative photosynthetic production and water loss of crops. Six wheat genotypes were studied for 2 years with pot cultivation in rain-shelter. Among different genotypes, stomatal conductance ($g_s$) was significantly correlated with both $P_N$ and $Tr$. But for each genotype, though $g_s$ was significantly correlated with $P_N$ regardless of relative air humidity (RH) status and it was also significantly correlated with $Tr$ under lower RH (LRH, 15.4%) and moderate RH (MRH, 28.3%), it was not correlated with $Tr$ under higher RH (HRH, 36.7%) during the diurnal changes. The conditional correlation between $g_s$ and $Tr$ of wheat evoked new thinking on the relationships among $g_s$, $P_N$ and $Tr$. Path analysis was further carried out to clarify the correlations of $g_s$ with the four atmospheric factors, that of $Tr$ with $g_s$ and the four factors and the direct and indirect effects of the factors, during their diurnal dynamic variation. The effects of these factors on $g_s$ or $Tr$ were related to RH. All the four factors had a much higher correlation with $g_s$ under HRH than that under LRH and MRH. Air temperature ($T$) had a rather higher direct effect than RH and photosynthetically active radiation (PAR). Also, the other factors had a much higher indirect effect on $g_s$ through vapor pressure deficit (VPD) and T. Transpiration rate was highly correlated with $g_s$ under LRH and MRH, with $g_s$ having a higher direct effect on it. In comparison, $Tr$ was not correlated with $g_s$ under HRH but highly correlated with the atmospheric factors, with T, RH, and PAR having a higher indirect effect through VPD.

## INTRODUCTION

Water shortage has been the most significant factor restricting plant growth and crop productivity with a deepening water-resource crisis worldwide. Wheat (*Triticum aestivum* L.) is one of the main crops consumed by humans and is cultivated in different environments. Only less than 30% of the rainfall occurs during the wheat growing season in the North China Plain, the main wheat production region of China, which meets only about 25–40% of the water requirements of wheat. As a result, more than 70% of the irrigation water is used for winter wheat (*Mei et al., 2013*). Irrigation usage for wheat threatens the sustainability of the groundwater resource (*Zhang, Pei & Hu, 2003*). Therefore, how to conserve soil water to enable sustainable crop production and maximize harvest of fields is becoming a main goal for many efforts of agriculture researchers.

Stomata, the gateway through which vapor and $CO_2$ pass, play an important role in regulating both photosynthesis and transpiration. Low stomatal conductance ($g_s$) results in low net photosynthesis rate ($P_N$) by restricting $CO_2$ uptake (*Farquhar & Sharkey, 1982*), while high $g_s$ benefits higher $P_N$ but at a greater expense of water loss via transpiration (*Lawson & Blatt, 2014*). The ability of stomata to exert rapid control of their aperture to minimize water loss while maintaining $CO_2$ uptake is one of the primary evolutionary mechanisms that has allowed terrestrial plants to survive and spread in an otherwise desiccating atmosphere (*Hetherington & Woodward, 2003*). A good understanding of the response and behavior of stomata and transpiration in winter wheat is in urgent need of water saving through the pathway of cultivar adoption and agricultural practices. Stomata adjusts aperture in response to diverse external stimuli, such as vapor pressure deficit (VPD) (*Devi, Sinclair & Vadez, 2010*; *Leonardi, Guichard & Bertin, 2000*), relative air humidity (RH) (*Bakker, 1991a*, *1991b*; *Merilo et al., 2018*; *Suzuki et al., 2015*; *Talbott, Rahveh & Zeiger, 2003*), soil moisture (*Belko et al., 2012*; *Kholová et al., 2010a*, *2010b*), air temperature (*T*) (*Haque et al., 2017*; *Hetherington & Woodward, 2003*), photosynthetic photon flux density (PPFD) (*McAusland et al., 2016*), and $CO_2$ (*Yoshimoto, Oue & Kobayashi, 2005*). These atmospheric factors continually vary at diurnal and seasonal rhythms (*Assmann & Wang, 2001*). Besides, the circadian clock was reported to control $g_s$ responses partly at least over the diurnal period (*Dodd et al., 2005*; *Hassidim et al., 2017*), with phase of the circadian clock adjusting to environmental cues, such as *T* and PPFD, etc. (*De Dios et al., 2016*; *Yin & Johnson, 2000*). The majority of these studies were conducted with controlled experiments. Few studies were performed utilizing different gradients of natural conditions, which may be largely different from those carried out in climate chambers or green houses. Besides, these researches seldom laid stress on how the influencing factors affect $g_s$ during the diurnal dynamic change.

Our previous study found that RH played an important role in determining the diurnal $g_s$ pattern of wheat. All tested genotypes under lower RH (LRH, 15.7%) and most genotypes under higher RH (HRH, 40.7%) displayed a gradual decline pattern from

Table 1 Genotype names and associated details.

| Genotype | Year of release | Breeding place | Pedigree |
| --- | --- | --- | --- |
| Jing 411 | 1993 | Beijing | Fengkang 2/Changfeng 1 |
| 12 Song 1 | Line | Henan | LK 906/Yan 7961 |
| Jinmai 47 | 1998 | Shanxi | 12057//Han 522/K37-20 |
| Lankaoaizao 8 | 2003 | Henan | Lankao 84(184)1/Lankao 90 |
| Chang 6878 | 2003 | Shanxi | Linhan 5175/Jinmai 63 |
| Zhoumai 18 | 2005 | Henan | Neixiang 185/Zhoumai 9 |

morning through the afternoon. All genotypes presented a single-peak curve pattern under moderate RH (MRH, 28.3%), but the peak time differed among genotypes (*Zhang et al., 2019*). Based on the previous research, this study aims at (1) clarifying how the influencing factors contribute to the diurnal variation of $g_s$ along with transpiration rate ($Tr$) of wheat under different RH and soil moisture regimes; (2) making clear how $P_N$ and $Tr$ of wheat are correlated with $g_s$ during the diurnal change under different regimes of soil moisture and RH. A deep understanding of the stomata and transpiration traits of wheat, the influencing factors and their effect on $P_N$ and $Tr$ under different conditions have great implications for saving water through cultivars and agricultural practices.

# MATERIALS AND METHODS

## Plant materials

Six winter wheat genotypes, Jinmai 47, Chang 6878, Jing 411, Zhoumai 18, Lankaoaizao 8 and 12 Song 1 were used as materials, with the names and associated details being listed in Table 1. The six genotypes were selected from wheat germplasm nursery affiliated to Luoyang Academy of Agriculture and Forestry, located in Henan Province, China. In April 2016, the jointing stage of winter wheat, stomatal conductance was measured in three replications of different genotypes in the nursery under rain-fed and well-watered conditions. According to the data collected from 9:00 to 11:00 AM, six genotypes, which were in a wide spectrum of $g_s$ from very low to rather high value in the two water conditions, were tentatively selected as materials.

## Growth conditions

The experiments were carried out at Shunyi Scientific Experimental Station, Institute of Environment and Sustainable Development in Agriculture, Chinese Academy of Agricultural Sciences, Beijing, China (40°N and 116°E, altitude 34 m) in 2017–2018 growing seasons of winter wheat. The precipitation and air temperature of every month during the growing season are presented in Fig. 1. Pot cultivation was adopted in this experiment. Polyvinyl chloride pots were 30 cm depth and 35 cm in diameter, with a drainage hole on the bottom. The pots were filled with 16 kg plow layer soil, which was sieved through a 5 mm sieve and then fully mixed. The soil nutrients were determined as 0.109 g/kg of total nitrogen, 14.4 g/kg of organic matter, 24.5 mg/kg of available phosphorus, 106 mg/kg of available potassium and soil pH was 7.7.

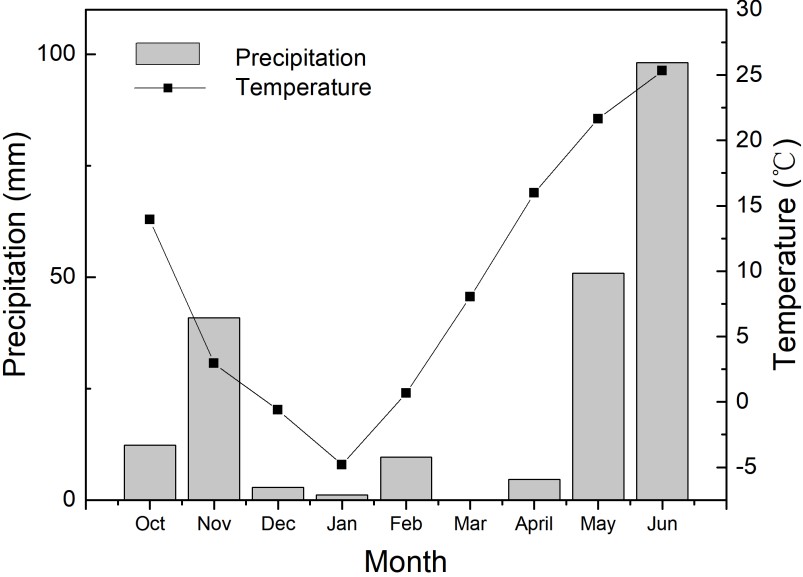

**Figure 1 The environmental condition during the 2017–2018 wheat growing season.** The data are the total precipitation and the average values of air temperature in each month.

For each genotype, six pots were planted for two water treatments and three replications. A total of 20 wheat seeds were sown in each pot on 5 October 2017. A total of 13 uniform seedlings were kept for overwintering, with the slender ones being removed on 30 days after emergence. After recovering in the spring, 10 uniform seedlings were eventually selected and kept as materials, with the unqualified ones being eliminated.

## Air humidity and soil water treatments

The treatments were different regimes of air humidity and soil water. The environmental RH gradient was taken as air humidity treatments. April 13, 28 and May 6, 2018, which were LRH (15.4%), MRH (28.3%) and HRH (36.7%) respectively, but similar in T and photosynthetically active radiation (PAR) were specifically selected. The specific diurnal meteorological conditions of the 3 days were shown in Fig. 2. Two water treatments were arranged: well-watered (WW) and drought stress (DS) conditions, with soil water content being 75% and 50% of field water holding capacity (FC), respectively. Water withholding for the two treatments was conducted by the weighing method. The plants were watered daily to restore the initial soil water content, between 1 and 2 h after sunset. Water withholding was kept for 30 days from April 6 to May 6. During water treatment periods, rain-shelter was used to prevent the rain on rainy days and was kept rolled up on sunny days to allow the crops to grow in the open-air conditions.

## Leaf gas exchange measurements

The youngest fully expanded leaves of the main tiller were measured for measuring the diurnal variation of gas exchange parameters under LRH, MRH and HRH from 8:00 to 18:00 with the 2 h interval, utilizing a Licor-6400 portable infrared gas analyzer (LI-COR Inc., Lincoln, NE, USA). For each replication, the leaves of three plants were measured and

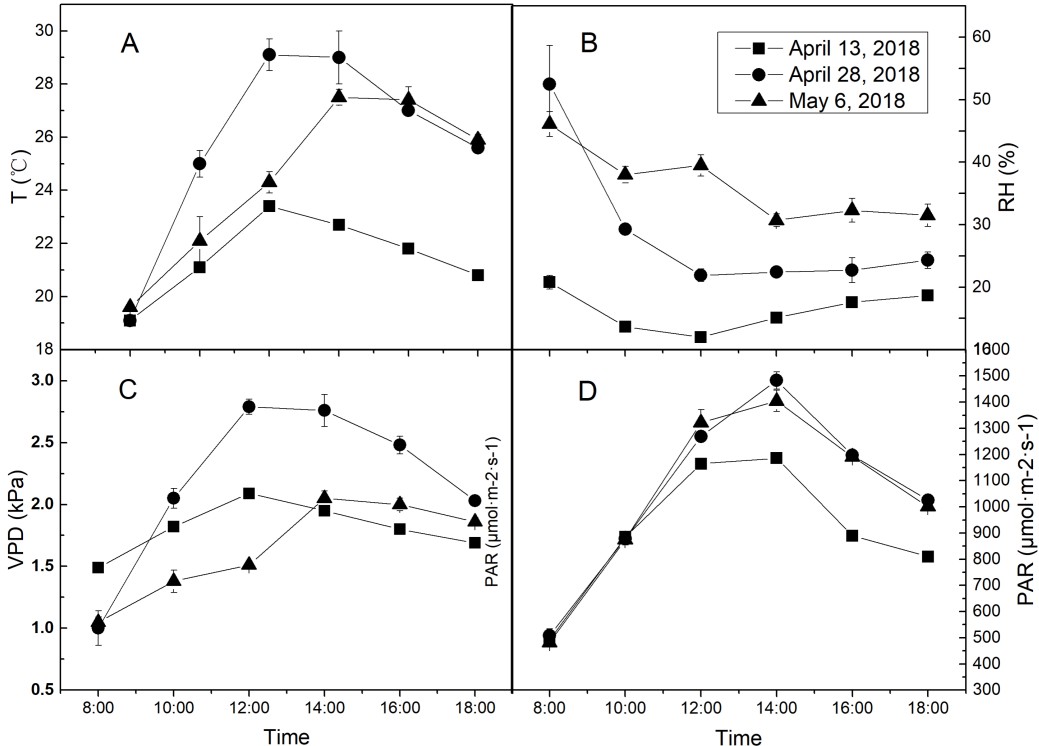

**Figure 2 The diurnal meteorological conditions of measurement days.** (A) Diurnal temperature (T) condition of each measurement day. (B) Diurnal relative air humidity (RH) condition of each measurement day. (C) Diurnal vapor pressure deficit (VPD) condition of each measurement day. (D) Diurnal photosynthetically active radiation (PAR) condition of each measurement day. Values represent means ± standard errors (*n* = 3).

averaged as the value for each time span. With one measurement, three parameters concerned, including $g_s$, $P_N$, $Tr$, were obtained in the meanwhile. To eliminate the disturbance to PAR from cloud cover and shading of neighboring leaves, PAR was set as the average value of 5 days in April or May for each time span. And chamber T was set as the same as the air temperature outside the chamber. RH could not be set for Li-cor-6400 gas analyzer, RH of the air flowing into the chamber was as same as the air RH. At the same time with diurnal measurement of gas exchange parameters, four atmospheric factors were also determined. Air temperature and RH were monitored once per minute by an automated temperature and relative humidity system (TH12R, Miaoxin, CN). VPD was calculated by T and RH as follows.

$$VPD = (1 - RH) \times 0.6108 \times e^{\frac{17.27 \times T}{T + 273.3}}$$

For investigating the relationships between $g_s$ with $P_N$ and $Tr$ across genotypes under identical soil water and meteorological conditions, leaf gas exchange parameters were measured with six different genotypes under two water treatments at 9:30–11:00 AM on April 16, 29 and May 5, 2018, the LRH (16.4%), MRH (25.9%) and HRH (37.8%) day respectively. The same gas analyzer described above were used, with the leaf chamber conditions being set at the same temperature of 25 °C, PAR of 1,000 μmol·m$^{-2}$·s$^{-1}$.

## Statistical analysis

The collected data were statistically analyzed by SAS software (SAS 9.4, Cary, NC, USA). A Pearson correlation analysis was used to assess correlations between parameters. Significance was considered at $P < 0.05$ and $0.01$. Data were presented as means ± standard errors ($n = 3$).

## RESULTS

### Main meteorological factors responsible for the diurnal variation of stomatal conductance

The four meteorological factors, T, RH, VPD and PAR interact and associate with each other. Each factor might directly affect $g_s$, also indirectly affect $g_s$ through other factors. Path analysis was carried out to clarify the correlations of four terms with $g_s$ and their direct and indirect effects on $g_s$ during diurnal dynamic variation. Table 2 showed how $g_s$ related to diurnally varying atmospheric factors depended on moisture regimes. All the four factors had a much higher correlation with $g_s$ under HRH than that under LRH and MRH. The correlation between $g_s$ and the four factors tended to be higher under WW than that under DS when RH was the same level. Of the four factors, $g_s$ was significantly correlated with T and PAR under all the regimes except DS + MRH. All the four factors were not correlated with $g_s$ under DS + MRH. Temperature had a rather higher direct effect than RH and PAR on $g_s$. And other factors had a much higher indirect effect on $g_s$ through VPD and T on $g_s$.

### Main factors responsible for the diurnal variation of transpiration rate

Stomatal conductance and the four main meteorological factors T, RH, VPD and PAR were closely related to each other. Path analysis was conducted to clarify the correlations between $Tr$ and the five factors, the direct effect and indirect effect through other factors on $Tr$ during their diurnal dynamic variation (Table 3).

Under LRH, $Tr$ was highly correlated with $g_s$ under both WW and DS, with higher direct effect coefficient (DPC = 0.7346) and a higher negative indirect effect through VPD and a positive indirect effect through T under WW, with higher direct effect (DPC = 0.8225) and a weak indirect effect through other factors under DS. Transpiration rate was also significantly correlated with T, RH and VPD under WW. The temperature had a higher positive direct effect and a higher negative indirect effect through VPD on $Tr$. RH had a relatively lower direct effect and a higher negative indirect effect through T and a higher positive indirect effect through VPD on $Tr$. VPD had a higher direct effect and a higher indirect effect through T on $Tr$. All the four meteorological factors had a lower indirect effect through $g_s$ on $Tr$.

Under MRH, $Tr$ had an extremely significant correlation with all the five factors, with the correlation between $Tr$ and $g_s$ being much higher, that between $Tr$ and RH being negative, and that between $Tr$ and T, VPD and PAR being positive. Stomatal conductance had a rather higher direct effect than the indirect effect through any other factors

Table 2 **Path analysis of stomatal conductance with meteorological factors during diurnal dynamic variation.** The measurement dates and diurnal meteorological conditions are shown in Fig. 2. $T$, air temperature; RH, relative air humidity; VPD, vapor pressure deficit; PAR, photosynthetically active radiation; WW, soil well-watered; DS, soil drought stress; LRH, lower relative air humidity; MRH, moderate relative air humidity; HRH, higher relative air humidity. The daily average relative air humidity of LRH, MRH and HRH were 15.4%, 28.3% and 36.7% respectively.

| Regimes | Independent variable | Direct path coefficient | Indirect path coefficient | | | | Simple correlation coefficient |
|---|---|---|---|---|---|---|---|
| | | | $T$ | RH | VPD | PAR | |
| WW LRH | $T$ | 1.8076 | | 1.1045 | −2.3004 | −0.3287 | 0.2830** |
| | RH | −1.2985 | −1.5376 | | 2.1758 | 0.2603 | −0.3999** |
| | VPD | −2.3397 | 1.7773 | 1.2075 | | −0.3159 | 0.3291** |
| | PAR | −0.3435 | 1.7297 | 0.984 | −2.1519 | | 0.2183* |
| DS LRH | $T$ | −7.4551** | | −1.802 | 8.8615 | 0.1602 | −0.2355* |
| | RH | 2.1159** | 6.3492 | | −8.3855 | −0.127 | −0.0474 |
| | VPD | 9.0120** | −7.3306 | −1.9688 | | 0.154 | −0.1335 |
| | PAR | 0.1675 | −7.1306 | −1.6041 | 8.2849 | | −0.2824** |
| WW MRH | $T$ | 0.0172 | | −0.3842 | 0.6178 | 0.0045 | 0.2553* |
| | RH | 0.4345 | −0.0152 | | −0.5996 | −0.0044 | −0.1847 |
| | VPD | 0.6316 | 0.0168 | −0.4125 | | 0.0047 | 0.2406* |
| | PAR | 0.0051 | 0.0153 | −0.3711 | 0.5763 | | 0.2257* |
| DS MRH | $T$ | −1.6275* | | −0.4768 | 2.0112 | 0.1741 | 0.081 |
| | RH | 0.529 | 1.4669 | | −1.9455 | −0.1709 | −0.1205 |
| | VPD | 2.0436 | −1.6017 | −0.5036 | | 0.1789 | 0.1172 |
| | PAR | 0.1994 | −1.4211 | −0.4535 | 1.8332 | | 0.158 |
| WW HRH | $T$ | −0.3377 | | 0.1237 | −0.4543 | −0.0818 | −0.7502** |
| | RH | −0.1346 | 0.3103 | | 0.4425 | 0.0776 | 0.6959** |
| | VPD | −0.4605 | −0.3332 | 0.1294 | | −0.0757 | −0.7400** |
| | PAR | −0.1288 | −0.2144 | 0.0812 | −0.2705 | | −0.5325** |
| DS HRH | $T$ | 2.0471 | | 1.082 | −3.3811 | −0.2891 | −0.5411** |
| | RH | −1.2121 | −1.8274 | | 3.3177 | 0.2236 | 0.5018** |
| | VPD | −3.4552 | 2.0031 | 1.1639 | | −0.2514 | −0.5396** |
| | PAR | −0.4298* | 1.3769 | 0.6306 | −2.0211 | | −0.4433** |

**Notes:**
* Significant difference at $p < 0.05$.
** Significant difference at $p < 0.01$.

on $Tr$. Of the four meteorological factors, VPD had the highest direct effect and T, RH and PAR had a higher indirect effect through VPD on $Tr$. The indirect effect of four meteorological factors through $g_s$ on $Tr$ was rather smaller.

Under HRH, $Tr$ was not correlated with $g_s$ but significantly correlated with the four meteorological factors except PAR under WW. VPD had the largest direct effect, while PAR had the smallest direct effect on $Tr$ under DS. The indirect effect of T, RH and PAR on $Tr$ through VPD was the largest, with that of RH being negative and that of T and PAR being positive. The indirect effect through $g_s$ of the four meteorological factors on $Tr$ was rather smaller.

Table 3 **Path analysis of transpiration rate with stomatal conductance and metorological factors during diurnal dynamic variation.** The measurement dates and diurnal meteorological conditions are shown in Fig. 2. $g_s$, stomatal conductance; $T$, air temperature; RH, relative air humidity; VPD, vapor pressure deficit; PAR, photosynthetically active radiation; WW, soil well-watered; DS, soil drought stress; LRH, low relative humidity; MRH, moderate relative humidity; HRH, high relative humidity. The daily average relative air humidity of LRH, MRH and HRH were 15.4%, 28.3% and 36.7% respectively.

| Regimes | Independent variable | Direct path coefficient | Indirect path coefficient | | | | | Simple correlation coefficient |
|---|---|---|---|---|---|---|---|---|
| | | | $g_s$ | $T$ | RH | VPD | PAR | |
| WW | $g_s$ | 0.7346** | | 0.9623 | 0.3766 | −1.1614 | −0.1552 | 0.7569** |
| LRH | $T$ | 3.4006** | 0.2079 | | 0.8009 | −3.4695 | −0.6801 | 0.2598* |
| | RH | −0.9416 | −0.2938 | −2.8927 | | 3.2816 | 0.5386 | −0.3079** |
| | VPD | −3.5288* | 0.2418 | 3.3435 | 0.8756 | | −0.6536 | 0.2785** |
| | PAR | −0.7107** | 0.1604 | 3.254 | 0.7135 | −3.2454 | | 0.1718 |
| DS | $g_s$ | 0.8225** | | 0.0305 | −0.0303 | −0.1454 | 0.0678 | 0.7451** |
| LRH | $T$ | −0.1296 | −0.1937 | | −0.5435 | 1.071 | −0.2296 | −0.0253 |
| | RH | 0.6381 | −0.039 | 0.1103 | | −1.0134 | 0.182 | −0.122 |
| | VPD | 1.0891 | −0.1098 | −0.1274 | −0.5938 | | −0.2206 | 0.0376 |
| | PAR | −0.24 | −0.2323 | −0.1239 | −0.4838 | 1.0013 | | −0.0787 |
| WW | $g_s$ | 0.7465** | | 0.0179 | 0.1135 | −0.1575 | 0.0687 | 0.7890** |
| MRH | $T$ | 0.0703 | 0.1906 | | 0.5432 | −0.6404 | 0.2707 | 0.4345** |
| | RH | −0.6143* | −0.1379 | −0.0621 | | 0.6215 | −0.2601 | −0.4529** |
| | VPD | −0.6547 | 0.1796 | 0.0687 | 0.5832 | | 0.2779 | 0.4547** |
| | PAR | 0.3045* | 0.1684 | 0.0625 | 0.5247 | −0.5974 | | 0.4627** |
| DS | $g_s$ | 0.7503** | | 0.0547 | 0.0912 | −0.1643 | 0.0373 | 0.7692** |
| MRH | $T$ | 0.675 | 0.0608 | | 0.6819 | −1.3795 | 0.2062 | 0.2444* |
| | RH | −0.7566** | −0.0904 | −0.6084 | | 1.3344 | −0.2024 | −0.3234** |
| | VPD | −1.4017 | 0.0879 | 0.6643 | 0.7203 | | 0.2118 | 0.2826** |
| | PAR | 0.2361 | 0.1186 | 0.5894 | 0.6485 | −1.2574 | | 0.3353** |
| WW | $g_s$ | 0.5982** | | 0.2367 | 0.2545 | −0.9967 | −0.0338 | 0.0589 |
| HRH | $T$ | −0.3155 | −0.4488 | | −0.336 | 1.329 | 0.0403 | 0.2690* |
| | RH | 0.3658 | 0.4163 | 0.2899 | | −1.2945 | −0.0383 | −0.2609* |
| | VPD | 1.347 | −0.4427 | −0.3113 | −0.3515 | | 0.0373 | 0.2787** |
| | PAR | 0.0635 | −0.3186 | −0.2003 | −0.2205 | 0.7912 | | 0.1153 |
| DS | $g_s$ | 0.5807** | | 1.0599 | 0.5937 | −1.8809 | −0.2289 | 0.1245 |
| HRH | $T$ | −1.9589 | −0.3142 | | −1.0561 | 3.4108 | 0.3472 | 0.4289** |
| | RH | 1.183 | 0.2914 | 1.7487 | | −3.3469 | −0.2686 | −0.3923** |
| | VPD | 3.4856* | −0.3133 | −1.9169 | −1.136 | | 0.302 | 0.4214** |
| | PAR | 0.5162** | −0.2574 | −1.3176 | −0.6155 | 2.0388 | | 0.3645** |

**Notes:**
* Significant difference at $p < 0.05$.
** Significant difference at $p < 0.01$.

## Stomatal conductance bear no correlation with transpiration rate in the diurnal change under HRH

The meteorological factors, such as T, RH and VPD, vary with time diurnally. Meteorological conditions were considered to be identical for different genotypes under

**Table 4 Correlations of stomatal conductance ($g_s$) with net photosynthesis rate ($P_N$) and transpiration rate ($T_r$) among six genotypes.** The measurements were conducted with six genotypes during 9:30–11:00 at each measurement day. WW, well-watered; DS, drought stress; LRH, lower relative air humidity; MRH, moderate relative air humidity; HRH, higher relative air humidity. The daily average relative air humidity of LRH, MRH and HRH were 16.4%, 25.9% and 3 7.8% respectively.

| Group | WW | | DS | |
|---|---|---|---|---|
| | $g_s$-$P_N$ | $g_s$-$Tr$ | $g_s$-$P_N$ | $g_s$-$Tr$ |
| LRH | 0.9460** | 0.8968** | 0.9663** | 0.9820** |
| MRH | 0.9100** | 0.9376** | 0.9096** | 0.9837** |
| HRH | 0.9547** | 0.6741** | 0.8222** | 0.9491** |

Note:
** Significant difference at $p < 0.01$.

**Table 5 Correlations of stomatal conductance ($g_s$) with net photosynthesis rate ($P_N$) and transpiration rate ($Tr$) in diurnal dynamics of each genotype.** The measurement dates and diurnal meteorological conditions are shown in Fig. 2. LRH, lower relative air humidity; MRH, moderate relative air humidity; HRH, higher relative air humidity. The daily average relative air humidity of LRH, MRH and HRH were 15.4%, 28.3% and 36.7% respectively.

| Genotype | LRH | | MRH | | HRH | |
|---|---|---|---|---|---|---|
| | $g_s$-$P_N$ | $g_s$-$Tr$ | $g_s$-$P_N$ | $g_s$-$Tr$ | $g_s$-$P_N$ | $g_s$-$Tr$ |
| 12 Song | 0.9726** | 0.9441** | 0.7655** | 0.6879** | 0.8910** | 0.3538 |
| Lankaoaizao 8 | 0.7316** | 0.7537** | 0.8859** | 0.8071** | 0.8437** | 0.2141 |
| Jinmai 47 | 0.8771** | 0.9019** | 0.8042** | 0.8425** | 0.7184** | 0.2127 |
| Chang 6878 | 0.8233** | 0.7739** | 0.8274** | 0.8633** | 0.7666** | 0.345 |
| Jing 411 | 0.9240** | 0.8909** | 0.5985** | 0.6794** | 0.8501** | 0.2251 |
| Zhoumai 18 | 0.8453** | 0.8514** | 0.9162** | 0.9560** | 0.8163** | 0.1919 |

Note:
** Significant difference at $p < 0.01$.

the same measurement time span (9:30–11:00 AM). Across different genotypes, $g_s$ had extremely significant correlations with $P_N$ and $Tr$ under identical meteorological conditions and soil water status (Table 4). The correlations between $g_s$ and $Tr$ under WW were lower than those under DS in the three different RH conditions. For the case of LRH, the correlations under WW were lower than those under DS. While for MRH and HRH, the correlation between $g_s$ and $P_N$ was higher under WW than that under DS. On the other hand, the correlations among $g_s$ with $P_N$ and $Tr$ in the diurnal dynamics were investigated with six genotypes individually under WW and DS conditions (Table 5). There were the same results in WW and DS whether respectively or collectively. Stomatal conductance was closely correlated with $P_N$ and $Tr$ under LRH and MRH. Interestingly, though $g_s$ was still closely correlated with $P_N$, it did not significantly correlate with $Tr$ for each genotype under HRH in the diurnal variation. This proved the high dependency between $g_s$ and $P_N$, also revealed the minor role of $g_s$ on determining $Tr$ diurnal variation under HRH. It might be other factors, mainly the meteorological factors, that predominantly control $Tr$ in the diurnal variation in such case.

## DISCUSSION

### Factors influencing the diurnal variation of stomatal conductance and transpiration rate

Stomata adjust aperture in response to environmental factors, such as soil water stress and atmospheric factors, as reported by many studies (*Xu et al., 2016*; *Hernandez-Santana et al., 2016*). The present study aimed at the relations between $g_s$ of wheat with the influencing factors during the diurnal dynamic variation and found that the relations was associated closely with RH. Stomatal conductance was not correlated with RH and VPD under LRH + DS and MRH + DS regimes, but highly correlated with the two factors under HRH. Moreover, stomatal conductance displayed a higher correlation with the influencing factors under WW than under DS. Thus, it seems that only when $g_s$ of wheat was relatively higher in the case of high air RH and soil moisture, it was closely related to atmospheric factors during the diurnal variation. The reduced stomatal aperture could not respond to diurnally varying factors sensitively under dry soil and air moisture, which explain the rather lower correlation between them. Stomatal conductance of wheat was significantly correlated with PAR in most of the regimes. *Sabir & Yazar (2015)* found that $g_s$ was better correlated with PAR for all the tested grapevines cultivars than the other measured meteorological parameters, including RH, T etc. This is consistent with the case under DS + LRH and WW + MRH in our study, implying that sunlight had a significant effect on stomatal diurnal response under the two regimes.

Transpiration rate is controlled by the plant itself through stomatal adjustment on one hand. As a passive diffusion process of water, it was also regulated by surrounding factors, such as T, RH and solar radiation intensity (*Yang et al., 2012*). Which factors mainly restrict $Tr$ during the diurnal variation depended on RH, as the present study showed. Higher correlation between $g_s$ and $Tr$ of wheat existed under LRH and MRH, but no correlation under HRH. This might imply that increased $g_s$ under HRH was not restricting factors for $Tr$, instead, atmospheric factors restricted $Tr$ in this case. While reduced $g_s$ became the main limiting factor for $Tr$ under lower RH. VPD had the highest direct effect of the four atmospheric factors and other factors had the highest indirect effect through VPD under almost all the regimes.

### Correlations between stomatal conductance with transpiration rate and photosynthesis rate in the diurnal dynamics

Across different genotypes under identical meteorological conditions and soil water status, extremely significant correlations existed between $g_s$ with $P_N$ and $Tr$, as reported by some previous studies (*Wong, Cowan & Farquhar, 1979*; *McAusland et al., 2016*). This indicates the important contribution of cultivars with different stomatal traits to photosynthesis and transpiration and in turn to yield formation and water consumption, in the case of identical environmental conditions.

During the diurnal dynamics, $g_s$ was significantly correlated with $Tr$ in all the wheat genotypes tested under LRH and MRH, but not under HRH (Table 5), indicating the correlation between them depended on RH. As a physical process of water passive

diffusion, transpiration is not only controlled by the plant itself through stomatal adjustment but also driven by soil moisture and atmospheric factors. In some cases, $Tr$ is mainly regulated by RH or VPD, instead of $g_s$. *Devi, Sinclair & Vadez (2010)* found that $Tr$ increased over VPD rising, with a break point occurring in some genotypes above which there was little or no further increase in $Tr$ of peanut. Stomatal conductance declined with VPD increasing (*Leonardi, Guichard & Bertin, 2000*; *Talbott, Rahveh & Zeiger, 2003*), or RH declining (*Fanourakis et al., 2016*, *2019*), which implied that $Tr$ did not decline with $g_s$ reducing, but increased with VPD rising. And the occurrence of $Tr$ breakpoint might be due to that $g_s$ continuously declined and turned to be the restricting factor of $Tr$. Also, *Aliniaeifard & Van Meeteren (2016)* conducted an experiment with *Chrysanthemum morifolium* plants in the growth chamber. Similarly, they found higher $g_s$ but lower $Tr$ under low VPD (high RH) in comparison with growth under moderate VPD. Consistently, *Giday et al. (2015)* found cultivar differences in plant transpiration rate at high rRH were not related to genotypic variation in stomatal responsiveness. The transpiration rate did not increase with $g_s$ rising but lowered down with increased RH. These indicated that it was VPD but not $g_s$ that controlled $Tr$ in such cases. Whether $g_s$ is closely related to $Tr$ depends on if $g_s$ is restricting $Tr$ under the specific circumstance. In the present study, the correlation coefficients between $g_s$ and $Tr$ of wheat under LHR and HRH were contrasting (Table 5). This might be ascribed to the different limiting effects of $g_s$ on $Tr$. Stomatal conductance went down and turned to be the limiting factor of $Tr$ under LRH. While enhanced $g_s$ is no more a limiting factor of $Tr$ under HRH and weakly correlated with $Tr$. It might be either the regulating and limiting effect of $g_s$ to $Tr$ or the synchronous response of $g_s$ and $Tr$ to the atmospheric factors that result in the high correlation between $g_s$ and $Tr$ under MRH.

Stomatal conductance was highly correlated with $P_N$ of wheat in the diurnal dynamics for each genotype, also across different genotypes at identical environmental conditions, highlighting the strong dependency between photosynthesis and stomatal regulation. However, as reported, stomata did not respond to environmental cues with $P_N$ synchronously, but an order of magnitude slower than $P_N$ (*Lawson & Blatt, 2014*). The lag in stomatal behavior and the temporal disconnect between $P_N$ and $g_s$ challenge the notion that stomata adjust the aperture to regulate $P_N$. Moreover, *Mott (1988)* reported that it was $CO_2$ concentration inside the leaf (Ci) rather than that outside the leaf influenced stomatal aperture. Afterwards, series of studies carried out by *Roelfsema et al. (2002*, *2006)* found that red light-induced stomatal opening is mediated by the reduction of Ci which is in turn caused by the increased photosynthetic activity of mesophyll cell. A recent study in maize (*Zea mays* L.) identified two *Ca* genes which encode carbonic anhydrase mediated the response of plants to increased Ci (*Kolbe et al., 2018*). As an organism of high auto-regulation, plants respond to environmental cues positively within its adaptation limits. Under some specific circumstances, photosynthesis, the initiative anabolism process, might be capable of regulating $g_s$ according to its demand for $CO_2$ through affecting Ci, though $g_s$ has been proved to be a limiting factor of photosynthesis by most studies (*Carmo-Silva et al., 2012*; *Chastain et al., 2014*).

## CONCLUSIONS

Relative air humidity played an important role in affecting the correlation between $g_s$ with $P_N$ and $Tr$ of wheat during the diurnal variation. The transpiration rate was not significantly correlated with $g_s$ but mainly affected by the atmospheric factors under HRH. In particular, VPD had a rather higher direct and indirect effect on $Tr$. The notion that stomata continuously adjust aperture in response to environmental factors to optimize the tradeoff between photosynthesis and water loss was challenged. Thus, the definite interrelationship among $P_N$, $Tr$ and $g_s$ of wheat needs to be elucidated conditionally. In cases where $g_s$ is not the key influential factor for $Tr$, any efforts to artificially reduce $g_s$ of wheat may not contribute significantly to water saving, but pay the price of photosynthetic reduction and yield loss. But when meteorological and soil water conditions were identical, $g_s$ was significantly correlated with $Tr$ and $P_N$ across different genotypes. Thus, to select and adopt appropriate wheat cultivars with specific stomata traits is undoubtedly a good strategy for realizing water saving. Taking yield issue into account, genotype adoption has to compromise the requirement for yield and water saving, and needs to match the water availability of areas with stomatal sensitivity of wheat genotypes to water stress.

### Funding

This work was supported by a grant from The National Key Research and Development Program (2017YFD0201702). The funders had no role in study design, data collection and analysis, decision to publish, or preparation of the manuscript.

### Grant Disclosures

The following grant information was disclosed by the authors:
The National Key Research and Development Program: 2017YFD0201702.

### Competing Interests

The authors declare that they have no competing interests.

### Author Contributions

- Xinying Zhang conceived and designed the experiments, performed the experiments, analyzed the data, prepared figures and/or tables, authored or reviewed drafts of the paper, and approved the final draft.
- Xurong Mei conceived and designed the experiments, authored or reviewed drafts of the paper, and approved the final draft.
- Yajing Wang performed the experiments, analyzed the data, prepared figures and/or tables, and approved the final draft.
- Guirong Huang performed the experiments, prepared figures and/or tables, and approved the final draft.
- Fu Feng performed the experiments, prepared figures and/or tables, and approved the final draft.
- Xiaoying Liu conceived and designed the experiments, authored or reviewed drafts of the paper, and approved the final draft.
- Rui Guo conceived and designed the experiments, authored or reviewed drafts of the paper, and approved the final draft.
- Fengxue Gu conceived and designed the experiments, authored or reviewed drafts of the paper, and approved the final draft.
- Xin Hu conceived and designed the experiments, authored or reviewed drafts of the paper, and approved the final draft.
- Ziguang Yang conceived and designed the experiments, authored or reviewed drafts of the paper, and approved the final draft.
- Xiuli Zhong conceived and designed the experiments, authored or reviewed drafts of the paper, and approved the final draft.
- Yuzhong Li conceived and designed the experiments, authored or reviewed drafts of the paper, and approved the final draft.

## Data Availability

The raw measurements are available in the Supplemental Files.

## Supplemental Information

Supplemental information for this article can be found online at http://dx.doi.org/10.7717/peerj.8927#supplemental-information.

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
