# Peer review of "Stomatal conductance bears no correlation with transpiration rate in wheat during their diurnal variation under high air humidity"

_PeerJ, doi:10.7717/peerj.8927_

## Round 0.1 · original submission · Major Revisions

The reviewers found that your work has a considerable contribution to its field and provide new and interesting knowledge. However, I concur with them that there are many significant issues that need to be fixed before further considering the paper. The reviewers proposed a detailed list of changes that must be taken into account when submitting a revised version of the ms. In particular, I’d like to recall the attention on the following:

- Rev #1 noted that a lot of o work has to be done for establishing a different order when writing the paper. It needs to elucidate between hypothesis and results step by step.
- Rev #2 found many severe limitations to the experimental setting of the work, including the definition of environmental conditions the samples were submitted to.
- Rev #3 remarked that throughout the manuscript, there are sentences, that are not easy to understand, and the meaning of the sentence remains unclear. Also, they suggest adding some figures to the ms, for improving the quality of the paper.

Reviewer 1 ·

Basic reporting

I have had serious problems when reading the manuscript. I cannot really find a guideline in the presentation of results. After reading the paper, I am not really sure of the real goal of the research.
I encourage the authors to rewrite the manuscript with a deep change in the sequential presentation of arguments and results prior discuss them and propose a conclusi

Experimental design

I am concerned with the idea of LHR, MHR and HRH. In my experience, the stomata response is affected by the HR (or VPD) in the chamber of the gas exchange system. In fact, it is hard to elucidate the effect of environmental (I mean the atmospheric HR or VPD) and chamber HR/VPD in the stomatal response. Can you rescue the VPD values in your measurement? Most of the gas exchange systems allow to modufy or not the HR of the air inlet by using dessicators. However, the a`pporttation of water vapour by the leaf induces a new equilibrium.

Validity of the findings

I encourage the authors to reordinate the results in order to clarify tehe reading. Do authors consider the internal regulations of stomatal sensitivity to HR or is merely a matter of pressure gradient? I am not really sure of the depth of their finding due to difficulties when reading the paper. In some sentences of the paper, really evident statements are done...The results should be written according to the hypothesis that guide the research to highlight the main goals

Additional comments

I encourage the authors to establish a different order when writting the paper. It is need to elucidate between hypothesis and results step by step, as the current presentation does not allow to appreciate clearly the real goal of the research. I propose to check the validity of gas exchange data by analysisng the real conditions during the measurement. Please, also check the relationship between the highly correlated HR, VPD and air temperature and discuss the functional differences (specially among the two firsts) in terms of stomatal sensitivity. A priori, I am not able of elucidate the real difference, but it may be a real and inetresting goal of the research. Moreover, I miss some comments about the specific sensitivity to HR of stomata and influence on the process under study.

Reviewer 2 ·

Basic reporting

no comment

Experimental design

no comment

Validity of the findings

no comment

Additional comments

The relation between PN, Tr and gs is here dealt in six wheat genotypes. According to the reviewer’s opinion, the current manuscript suffers from the following severe limitations, ought to be dealt by the authors with due caution: (1) environmental conditions during plant growth are not reported; (2) treatments refer to short term changes in environmental variables (nowhere mentioned); (3) the RH range is too small (15–36%). Deserves to be noted that high RH in other studies is above 85%; (4) VPD cannot be compared with either T or RH, since VPD is calculated based on T and RH; (5) the effect of long-term soil water deficit is nowhere to be seen.

Lines 44, 177: what do you mean by uniform?
Lines 46, 95: correct term is relative air humidity (abbreviated as RH). Please correct it here, and elsewhere in the text.
Lines 47–49 & 95–97: report the values of RH. What do you mean by low, moderate and high RH? Please provide the values.
Lines 54–55: not correct to compare VPD with either T or RH, since VPD includes both T and RH. You may compare T with RH, but not VPD with these two environmental variables.
Line 113: Table 1 reports the genotypes, and not environmental conditions. Besides environmental conditions during measurement (reported in Table 2), it is of outmost interest to report the environmental conditions during the whole growth period. This is missing from the study!
Lines 154–156: Thus treatments are actually environmental conditions during growth. This is not reported in the abstract, results and discussion section! The way the paper is written, it gives the erroneous notion that the RH treatments refer to the growth period. However, what is really meant, and nowhere can be seen, is that short-term changes in environmental conditions. Please clarify this everywhere on the text.
Line 233: high RH in the literature is above 85%. In this study, high RH is only 36.7%. In the remaining literature 36.7% is rather low RH.
Line 235: soil moisture affects plant transpiration through regulation of stomatal opening.
Line 236: “Leaf transpiration rate during cultivation is dominated by RH, with gs exerting minor role” in Fanourakis D, Giday H, Hyldgaard B, Bouranis D, Körner O, Ottosen C-O (2019) Low air humidity during growth promotes stomatal closure ability in roses. European Journal of Horticultural Science 84, 245–252.
Line 243–245: This study deals with another issue, as compared to the one dealt in this study. For sure, mentioning the long-term effects (i.e., throughout growth) of high RH is well-deserved. Note that in this case high RH means higher than 85%. Plants grown under high (≥85%) RH develop stomata that are not responsive to closing stimuli. In the growth environment, transpiration rate is dominated by factors other than gs, which is continuously high. Please check and report the following reviews dealing with this subject: Fanourakis et al 2016 (Journal of Plant Physiology 207, 51–60) & Fanourakis et al. 2015 (Acta Horticulturae 1064, 195–204). These reviews present the role of stomatal opening in determining transpiration in the growth environment, as well as in environments with low (40%) RH.
Line 264–269: Did the authors assess Ci? Why is this relevant here?
Discussion: the effect of soil water deficit is missing!!

Reviewer 3 ·

Basic reporting

The study investigates the correlation of stomatal conductance with photosynthesis rate and transpiration rate under varying air relative humidity in wheat. It is concluded, that under low and moderate air relative humidity, stomatal conductance and transpiration rate are significantly correlated, but under high relative humidity no correlation occurs and the variation in transpiration rate under high relative humidity is rather explained by varying environmental factors.
The main strength of the study is that it investigates an interesting topic concerning sustainable crop production. The results of this study have a considerable contribution to its field and provide new and interesting knowledge. Sufficient field background is provided and it can be understood, how the work fits into the broader field of knowledge.
Relevant prior literature is referenced, although the referencing and the reference list does not entirely comply with the journals’ style. For example, there are missing publication years on lines 237, 243, 263, 265. The reference list is not uniform, some journal names are in full, some are abbreviated, in some article titles each word is written with a capital letter. One reference (Mott on line 263) is missing from the reference list. The paragraph on lines 94-101 and sentence on line 241 needs a citation.
The English language used in the manuscript is not clear and unambiguous. Throughout the manuscript, there are sentences, that are not easy to understand and the meaning of the sentence remains unclear. The manuscript should undergo a thorough and professional proofreading. A few examples of non-professional English and hard to understand sentences are on lines 59 (“higher effected”), 80-82, 122-127, 133-135, 138-149 (mixed up and incomplete sentences, repetitions).
There are no Figures in the manuscript. I strongly recommend to add graphs showing the diurnal variation of temperature, air relative humidity, VPD, stomatal conductance, photosynthesis rate and transpiration rate. Adding these figures would greatly improve the manuscripts’ quality.
Tables are incorrectly cited. For example, it is referred to Table 1 on line 113, but in fact the data are in Table 2. In my opinion, there should not be any citations to Tables in the Statistical analysis section (lines 168-172), all the more the citations are incorrect (citation to Table 4 refers to Table 3; Table 5 to Table 4).
In Table 3, it is unclear, how many genotypes were analyzed – in the title it is said fifteen but in the legend, it is said six. No units are given in Table 3, 4, 5 and 6. In Tables 3 and 4, an abbreviation “A” is used, but it is not defined (should there be PN instead?).

Experimental design

The submission is original primary research within Aims and Scope of the journal, but research questions are not defined. The knowledge gap is somewhat identified on lines 70-71, 80-82 and 102-105, but the aim of the study and hypotheses are absent. The last two sentences of Introduction (lines 105-107) should be omitted, as they give a conclusion of the findings in the current study and should be placed in the Conclusions.
In general, methods are described sufficiently, if the language usage problems are not taken into account. Some questions still arise. The altitude of the experimental site is important to know, so it should be added (line 113). It is not stated, how many plants were measured for stomatal conductance, in order to identify genotypes with a wide spectrum of gs values (line 127). Due to the language usage problems, it is not clear, how many replications were used in the experiment. On lines 162-165, the models and the producers of the used apparatuses are not given. In the statistical analysis section, it is not stated, whether the normality and homogeneity of the data are checked.

Validity of the findings

No comment.

Additional comments

The sentence on lines 180-182 does not belong to Results but should rather be placed in the Discussion.

---

## Round 0.2 · Minor Revisions

Reviewer 2 asks for further minor changes regarding the RH effects and suggests some literature that may be considered to strengthen the discussion.

Reviewer 2 ·

Basic reporting

reviewer's comments were adequately dealt

Experimental design

reviewer's comments were adequately dealt

Validity of the findings

reviewer's comments were adequately dealt

Additional comments

The authors dealt with my comments respectfully.
I just realized that one point is still missing in the discussion and ought to be incorporated.
RH was found to be the main driver of environmental differences.
Having said that, it must be highlighted that the RH effect on transpiration rate also stimulates differences in leaf temperature (Giday et al. (2015) Cultivar differences in plant transpiration rate at high relative air humidity are not related to genotypic variation in stomatal responsiveness. Acta Horticulturae 1064, 99–106.) Please integrate this at your discussion to further strengthen the noted RH effects.

---

## Round 0.3 · Minor Revisions

After having considered the changes made to the previous version, I think that your manuscript steel needs some changes. One of the editors noted that there are some over-bearing statements when this study was only conducted on wheat. Likewise, perhaps it would be more useful to include some pedigree data in Table 1 to determine if there are any common parentage.

As a minor/major change, the paper still needs considerable editing for proper language use. Some examples are reported below:

'Greatly understanding the response of photosynthesis rate (P N ) and transpiration rate (Tr) to stomatal alteration during the diurnal variations are of importance for cumulative photosynthetic product and water loss of crops.' -> 'A good understanding of the response of photosynthesis rate (P N ) and transpiration rate (Tr) to stomatal alteration during the diurnal variations is important to cumulative photosynthetic product and water loss of crops.'

'The meteorological factors, such as T, RH, and VPD, etc., varies with time diurnally.' (...vary with...') - also, I do not see a definition for VPD until the third time it is used in the text (L82)

'Stomatal conductance highly correlated with...' ('...is highly...')

Please provide the requested changes in the revised version of your paper along with a rebuttal letter.

---

## Round 0.4 · accepted · Accept

Thank you for making the requested changes. I think that the paper has improved and it is now suitable for publication.